# Identification of a Novel Dehydrogenase from *Gluconobacter oxydans* for Degradation of Inhibitors Derived from Lignocellulosic Biomass

**Hongsen Zhang** [1], **Jiahui Jiang** [1], **Conghui Quan** [1], **Guizhong Zhao** [1], **Guotao Mao** [1], **Hui Xie** [1], **Fengqin Wang** [1], **Zhimin Wang** [2], **Jian Zhang** [3], **Pingping Zhou** [4,*] and **Andong Song** [1,*]

1   College of Life Sciences, Henan Agricultural University, Zhengzhou 450002, China; hszhang@henau.edu.cn (H.Z.); maoguotao@henau.edu.cn (G.M.); xiehui@henau.edu.cn (H.X.); w_fengqin@henau.edu.cn (F.W.)
2   College of Science, Henan Agricultural University, Zhengzhou 450002, China; gary1451@iccas.ac.cn
3   College of Bioengineering, East China University of Science and Technology, Shanghai 200237, China; jzhang@ecust.edu.cn
4   College of Food and Biology Engineering, Henan University of Animal Husbandry and Economy, Zhengzhou 450046, China
*   Correspondence: ppzhou@hnuahe.edu.cn (P.Z.); songandong@henau.edu.cn (A.S.)

**Abstract:** Inhibitors from lignocellulosic biomass have become the bottleneck of biorefinery development. *Gluconobacter oxydans* DSM2003 showed a high performance of inhibitors degradation, which had a short lag time in non-detoxified corn stover hydrolysate and could convert 90% of aldehyde inhibitors to weaker toxic acids. In this study, an aldehyde dehydrogenase gene *W826-RS0111485*, which plays an important function in the conversion of aldehyde inhibitors in *Gluconobacter oxydans* DSM2003, was identified. *W826-RS0111485* was found by protein profiling, then a series of enzymatic properties were determined and were heterologously expressed in *E. coli*. The results indicated that NADP is the most suitable cofactor of the enzyme when aldehyde inhibitor is the substrate, and it had the highest oxidation activity to furfural among several aldehyde inhibitors. Under the optimal reaction conditions (50 °C, pH 7.5), the *Km* and *Vmax* of the enzyme under furfural stress were 2.45 and 80.97, respectively, and the *Kcat* was 232.22 min$^{-1}$. The biodetoxification performance experiments showed that the recombinant *E. coli* containing the target gene completely converted 1 g/L furfural to furoic acid within 8 h, while the control *E. coli* only converted 18% furfural within 8 h. It was further demonstrated that *W826-RS0111485* played an important role in the detoxification of furfural. The mining of this inhibitor degradation gene could provide a theoretical basis for rational modification of industrial strains to enhance its capacity of inhibitor degradation in the future.

**Keywords:** inhibitors degradation; *Gluconobacter oxydans*; protein profiling; aldehyde dehydrogenase; furfural

## 1. Introduction

Lignocellulose as a renewable biomass resource has a wide range of sources and abundant reserves [1]. Lignocellulose has high antioxidant activity, but due to its complex structure, it is generally difficult to be directly used. Therefore, it is usually necessary to separate the biomass components by pretreatment, and then remove lignin, protect hemicellulose, and reduce the crystallinity of cellulose to make cellulose more easily converted to polysaccharides by enzymes [2]. Different pretreatment methods can produce different concentrations and different kinds of inhibitors, which affect the subsequent hydrolysis and fermentation efficiencies. Under harsh pretreatment conditions, cellulose, hemicellulose, and lignin are mainly degraded to produce three types of inhibitors: furan compounds (furfural, 5-hydroxymethyl furfural, etc.), phenolic compounds (vanillin, syringaldehyde, 4-hydroxybenzaldehyde, etc.), and weak acids (formic acid, acetic acid, etc.) [3–5]. Inhibitors

can create a hostile environment for fermentative microorganisms, extend the lag time, and lead to the loss of cell density and the reduction of the growth rate for fermentative microorganisms [6]. More seriously, these inhibitors can destroy the integrity of biological cell membranes, enter cells, affect intracellular energy balance, inhibit protein synthesis and the activity of key enzymes in the synthesis path of target products, and ultimately reduce product yield [7].

Biological detoxification refers to the use of microorganisms to degrade or convert the inhibitors from the pretreatment process of lignocellulose. Generally, they have the advantages of mild reaction conditions, no chemical additions, few side reactions, and low energy requirements [8,9]. The results of a recent study showed that biological detoxification with microorganisms is more effective than biological detoxification with adsorbents [10]. Most importantly, the wild type or after domesticated microorganisms are both used to biological detoxification, biological detoxification conditions similar to those of fermentation, so that the fermentation and biological detoxification process can be carried out at the same time, which can reduce a lot of by-product production and shorten the fermentation time [11,12].

The earliest microorganisms used for biological detoxification were *Filamentous fungi*. In 1997, it was first reported that *Trichoderma reesei* could be used to degrade the inhibitors (acetic acid, furfural, and 5-HMF) during the fermentation process [13]. In addition, many studies have shown that a variety of yeast and bacteria also have the ability of detoxification. So far, the 24 aldehyde reductases have identified from *S. cerevisiae*. Almost all of these dehydrogenases convert aldehyde inhibitors to weaker toxic alcohols, such as furfural to furfuryl alcohol [14,15]. Coincidentally, most bacteria covert aldehyde inhibitors to alcohols first, such as *Corynebacterium glutamicum*, *Zymomonas mobilis*, and so on [14,15]. However, *Gluconobacter oxydans* draws much attention because it can directly convert aldehyde inhibitors to acids [16]. Acids from aldehyde are more likely to enter the central metabolic pathway of the bacteria and be completely degraded into carbon dioxide and water, but alcohol is difficult to be completely degraded.

*G. oxydans* belongs to the acetobacter group (AAB) and is an aerobic Gram-negative bacteria [17]. Owing to the strong incomplete oxidation capacity of *G. oxydans* and the abundance of dehydrogenases in its cell membrane, most carbon sources can be incompletely oxidized and converted to acidic products, such as gluconic acid from glucose [18]. In previous studies on the production of lignocellulosic gluconic acid, *G. oxydans* showed a very short lag period in the corn stover hydrolysate, and it can directly convert more than 90% of aldehyde inhibitors into acids. DNA microarray technology has been used to discover genes related to the conversion of furan and phenolic inhibitors. For example, *W826-RS0110195*, *W826-RS0105635*, and *W826-RS0110240* may be involved in the conversion of aldehyde inhibitor to acid [16]. However, the dehydrogenases that play a role in the conversion of aldehyde inhibitors have not been identified.

In this study, an aldehyde dehydrogenase was identified from the mass spectrometry results of cell membrane-bound proteins in *G. oxydans* under inhibitor stress. The purified aldehyde dehydrogenase was purified, and its physiological characteristics were analyzed to obtain the optimal reaction temperature, pH, and the reaction product. The aldehyde dehydrogenase was cloned and expressed in *E. coli* and achieved satisfactory results.

## 2. Materials and Methods

### 2.1. Strains and Materials

Furfural, 5-hydroxymethylfurfural, vanillin, and 4-hydroxybenzaldehyde were purchased from Aladdin (95%).

*G. oxydans* DSM 2003 used in this study was purchased from German Collection of Microorganisms and Cell Cultures, Germany, Braunschweig. The strains were cultured in seed medium containing 80 g/L sorbitol, 20 g/L yeast extract, 0.5 g/L $MgCl_2 \cdot 7H_2O$, 1.5 g/L $KH_2PO_4$, and 1.5 g/L $(NH_4)_2SO_4$. The seed medium was sterilized at 115 °C for 15 min. The culture conditions were 30 °C and 220 rpm for 24 h, and the inoculum size was 10%.

## 2.2. Cloning, Expression, and Purification

The whole genome of *G. oxydans* DSM2003 was used as a template, the PCR primers were designed by Snapgene software, and the pET-28a(+) was digested with *Nde I* and *Hind III* (New England Biolabs, Beijing, China), respectively [19]. Following that, the target gene was connected to pET-28a(+) by homologous recombination, and the recombinant plasmid was transferred into the cloning vector DH5α. After verification, the recombinant vector was transformed into *E. coli* BL21(DE3) and cultured in Luria-Bertani (LB) medium supplemented with 25 mg/L kanamycin. When the culture reached OD600 of 0.6 to 0.8, 0.5 mM IPTG was added to induce expression of the target gene, and then the cells continued to grow at 16 °C for 16 h. Cells were harvested by centrifugation at $7000 \times g$ for 10 min and homogenized using a JN-Mini homogenizer (JNBio, Guangzhou, China). The target protein in the supernatant was purified using an Ni column. The purified target proteins were stored at −80 °C in 20 mM phosphate buffer (pH 7.4).

## 2.3. Enzyme Activity Assay

The enzyme activity was measured by the reduction of DCIP at 600 nm. The reaction system consisted of 0.3 mM PMS, 0.2 mM DCIP, 1 g/L inhibitor, 0.02 mM PBS buffer, and an appropriate amount of enzyme solution for 10 min. One unit of enzyme activity was defined as the amount of enzyme catalyzing the oxidation of 1 μmol of substrate per min, which was calculated using the millimolar extinction coefficient of DCIP of 13.2 at pH 6.5 and of 11.13 at pH 6.0 [20].

The NADP method was used to measure the enzyme kinetic parameters. The reaction system consisted of 1 mM NADP, 1 g/L inhibitor, 0.02 mM PBS buffer, and an appropriate amount of enzyme solution for 10 min. The absorbance value at 340 nm was measured by a microplate reader, and the enzyme activity was calculated by making a standard curve with different concentrations of NADPH. The unit of enzyme activity U was defined as one unit of enzyme activity required to catalyze the production of 1 mmol NADPH in 1 min [21].

## 2.4. Effects of Temperature and pH on the Activity and Stability

The reaction system consisted of 0.3 mM PMS, 0.2 mM DCIP, 1 g/L inhibitor, 0.02 mM buffer of different pH, and appropriate enzyme solution for 10 min. The absorbance value at 600 nm was measured by a microplate reader to quantitatively determine the REDOX reaction. The effect of pH on enzyme activity was determined with citrate-sodium citrate buffer (pH 4.0, 5.0), phosphate buffer (pH 5.5, 6.0, 6.5, 7.0, 7.5, 8.0), and Tris-HCl (pH 8.5, 9.0). To determine its pH stability, the purified enzyme was incubated with different pH buffers at 4 °C, and the residual enzyme activity was determined.

The effect of temperature (20 °C, 25 °C, 30 °C, 35 °C, 40 °C, 45 °C, 50 °C, 55 °C, 60 °C, 65 °C, 70 °C, 80 °C) on enzyme activity was determined separately in 20 mM phosphate buffer (pH 7.4) [19]. They were incubated at different temperatures for different times to determine the residual enzyme activity. The reaction system was described as above.

## 2.5. Inhibitor Analytical Methods

Furfural, furfuryl alcohol, and 2-furoic acid (furoic acid) were analyzed using reverse phase HPLC (UV detector, UltiMate 3000, Thermo, Waltham, MA, USA), equipped with an Agilent 5 TC-C18(2) column (Agilent, Santa Clara, CA, USA), using 10% methanol-acetonitrile: 0.1% formic acid = 1:4 at 1.0 mL/min, the column temperature of 35 °C, and the detection wavelength of 230 nm [16].

## 2.6. Protein Mass Spectrometry

Protein mass spectrometry was submitted to Shanghai Applied Protein Technology Co., Ltd. (Shanghai, China) The QE mass spectrometer was used to obtain tandem mass spectra by the principle of HCD (Higher-energy collisional dissociation). The detailed method used in this study was as follows: (1) Enzymolysis of protein samples. After reduction and alkylation, the protein samples were added with Trypsin, and the enzymolysis

was conducted at 37 °C for 20 h. The enzymatic hydrolysis products were freeze-dried after desalting, redissolved in 0.1% FA solution, and stored at −20 °C for future use. (2) Mass spectrometry. Liquid A was 0.1% formic acid aqueous solution, and liquid B was 0.1% formic acid acetonitrile aqueous solution (84% acetonitrile). After the column was balanced with 95% liquid A, samples were fed to the Trap column by an automatic sampler. (3) Mass spectrum data collection. Mass charge ratios of peptides and peptide fragments were collected as follows: 20 fragment maps (MS2 scan) were collected after each full scan. (4) Data analysis. The raw file of mass spectrometry test was retrieved from the corresponding database with Proteome Discoverer1.4 software (Thermo, Waltham, MA, USA), and finally the identified protein results were obtained.

## 3. Results and Discussion

### 3.1. Screening of Aldehyde Inhibitor Degradation Genes Derived from G. oxydans DSM 2003

The oxidoreductases in *G. oxydans* can be divided into membrane-bound dehydrogenases and soluble oxidoreductases in the cytoplasm according to the regional localization of enzymes in cells. The intracellular soluble dehydrogenase system is thought to be mainly responsible for the plateau phase of cell growth, while the membrane-bound dehydrogenase system is responsible for the rapid incomplete oxidation of many important substrates [22]. The membrane-bound proteins of *G. oxydans* DSM 2003 play an important role in the rapid oxidation of aldehyde inhibitors. In order to screen key genes that may play an important role in the rapid oxidation of aldehyde inhibitors, the membrane proteins of *G. oxydans* DSM 2003 were obtained by cell fragmentation and ultra-high speed centrifugation, then separated and purified from large to small by gel filtration chromatography column Superdex 200 Increase 10/300 GL (Figure 1). The results showed that membrane proteins were arranged from large to small from lane 1-4 to 2-2 (due to too many proteins on the cell membrane or some proteins being aggregated, the proteins in each lane were not strictly distributed from large to small). The activity of each fraction of proteins to four aldehyde inhibitors (1.0 g/L furfural, 1.0 g/L 5-hydroxymethylfurfural, 1.0 g/L vanillin, 1.0 g/L 4-hydroxybenzaldehyde) was detected and compared, and the REDOX indicator DCIP was used to represent the degree of REDOX reaction (Figure 2). As shown in Figure 2, the protein from collection tubes 1-6 to 2-2 showed higher activity to four aldehyde inhibitors, with more than 50% and 30% of the activity to furan aldehyde and phenolic aldehyde, respectively, compared with the whole cell membrane proteins. The proteins of the 2-6 collecting tube showed the highest activity against the four inhibitors, and the activities against furan aldehyde and phenolic aldehyde were still retain more than 90% and 70% compared with that of the whole cell membrane fraction, respectively. Combined with the electrophoresis results shown in Figure 1, the composition of proteins with sizes between 40 and 120 kDa (the part shown in red box in Figure 1) was significantly different between the higher activity 2-6 and the lower activity 1-4, and it was speculated that the active proteins were concentrated between 40 and 120 kDa. Part of the 40–120 kDa bands in "1-4" and "2-6" were cut off for protein profiling. A total of 45 proteins were screened, including dehydrogenases, oxidoreductases, and uncharacterized proteins. However, there is a gene with a high relative expression and expression of aldehyde dehydrogenase, A0A829WIG4, which was compared with the NCBI database. It was found that the gene number *W826-RS0111485* in *G. oxydans* DSM2003 was highly similar to it.

ExPASy online software was used to analyze *W826-RS0111485* with 463 amino acid residues, of which 255 were hydrophobic amino acids, and the content of hydrophobic amino acids was more than 55%. The overall average hydrophilic coefficient was −0.120. The relative molecular mass was 50.5 kDa, the theoretical isoelectric point was 5.69, and the extinction coefficient was 0.989. Online prediction via TMHMM and SignalP 5.0 website (accessed on 7 September 2022) showed that the protein had no transmembrane helix region, and the probability of no signal peptide was 0.9934. Therefore, it is speculated that the protein, although highly hydrophobic, does not belong to the transmembrane protein. During the experiment, a large amount of protein was detected in the cell membrane

fraction, which may be due to its large amount of hydrophobic amino acids, resulting in its presence near the cell membrane.

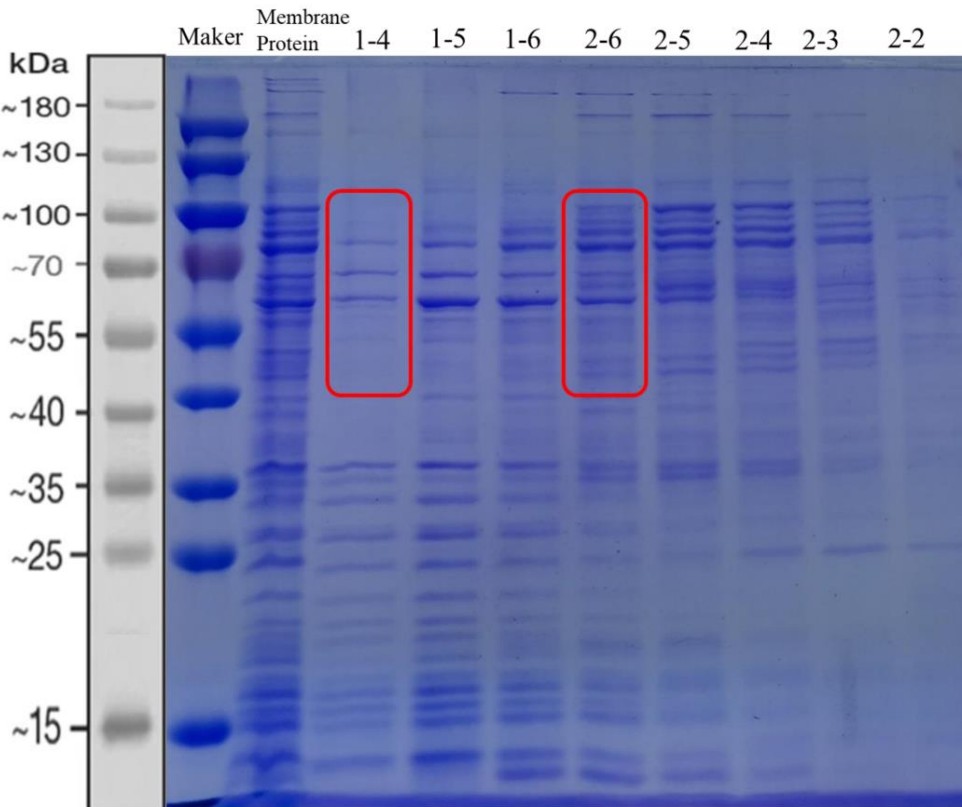

**Figure 1.** SDS-PAGE of proteins collected from each fraction by gel filtration chromatography. Lane 1 is the 180 kDa Protein Marker, lane 2 is the whole cell membrane proteins of *G. oxydans* DSM2003, lanes 3–10 are the proteins collected by gel filtration chromatography column Superdex 200 Increase 10/300 GL. The bands in the red box are the proteins analyzed by mass spectrometry in this study.

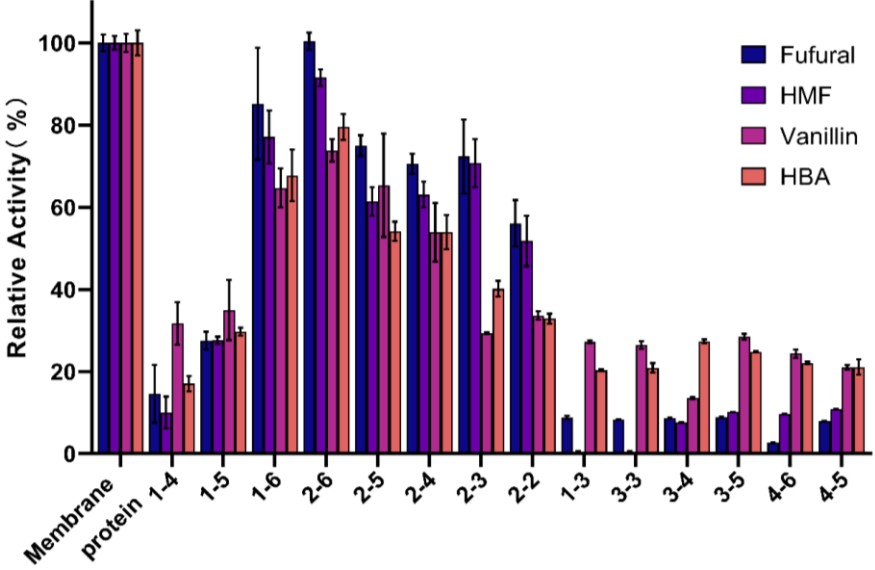

**Figure 2.** Comparison of the activity of each fraction of protein against four inhibitors (furfural, 5-hydroxymethylfurfural, vanillin, and 4-hydroxybenzaldehyde) collected by gel filtration chromatography. The concentration of inhibitor used was 1 g/L, and the REDOX indicator DCIP was used to indicate the degree of REDOX reaction (the viability of the membrane component was 100%).

The target gene sequence was searched in NCBI, and the whole genome of *G. oxydans* DSM2003 was used as a template. The PCR amplification primers were designed by Snapgene software, and the homology arm was added to the 5′ and 3′ end of the primer, respectively. After sequencing and verification, the plasmid was transferred into BL21 competent cell to induce expression at overnight low temperature. After the cells were broken by low temperature and ultra-high pressure, the resulting crude enzyme solution was purified by Ni column and concentrated by ultrafiltration to obtain a high concentration of pure enzyme for the following experiments (Figure 3).

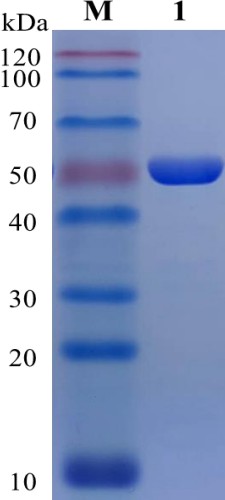

**Figure 3.** Protein purification. Lane M represents Maker at 120 kDa, and lane 1 represents the target protein after purification by the Ni column.

## 3.2. Effect of Cofactors on Enzymatic Activity of W826-RS0111485

So far, many important membrane-bound dehydrogenases in *G. oxydans* have been identified, such as membrane-bound glucose dehydrogenase [23], sorbitol dehydrogenase [24], alcohol dehydrogenase [25], and glycerol dehydrogenase [26], etc. Most of them use PQQ, FAD, and FMN as coenzymes or prosthetic groups [27]. In this study, the effect of different cofactors (NAD, NADP, PQQ, NADH, NADPH) on the activity of *W826-RS0111485* in the conversion of different aldehyde inhibitors was also studied (Figure 4). The results showed that NADP was the optimal cofactor of *W826-RS0111485*. The activity of *W826-RS0111485* against furfural, HMF, vanillin, and 4-hydroxybenzaldehyde was about 3000, 1000, 500, and 150 times higher than that without NADP. At the same time, when NAD was used as a cofactor, the activity of furfural was also significantly increased, which was 700 times higher than that of control. Therefore, the subsequent experiments to detect the optimal reaction conditions of the enzyme will be carried out with furfural as the substrate and NADP as the cofactor.

## 3.3. Effects of Temperature and pH on the Activity and Stability of W826-RS0111485

The environmental temperature has a great impact on the activity of the enzyme, and the appropriate temperature is more conducive to the maximum activity of the enzyme. The experimental results showed (Figure 5A) that the optimal reaction temperature of the enzyme was 50 °C, and the activity of the enzyme was still 90% at 45–60 °C, but at 65 °C the activity of the enzyme decreased to only 30%. In addition to properties of the enzyme itself, there is another conjecturing reason for why NADP would decompose at temperatures higher than 60 °C, resulting in lower reaction rate. The optimum reaction temperature 50 °C of the enzyme is higher than the optimum growth temperature 30 °C of *G. oxydans*. In fact, this situation has also occurred in several prokaryotes, such as membrane-bound dehydrogenases in *Methanosarcina mazei* [28] and glycerol kinase in *Flavobac-terium meningosepticum* [29]. However, its thermal stability was not outstanding

(Figure 5B). After incubation at 60 °C for 30 min, the enzyme activity was 5.10%. After incubation at 50 °C for 30 min and 1 h, 28.36% and 8.87% of the enzyme activity remained, respectively. After 8 h incubation at 30 °C, only about 70% of the enzyme activity remained. In the future, it is planned to improve the thermal stability of the enzyme by modifying the protein structure.

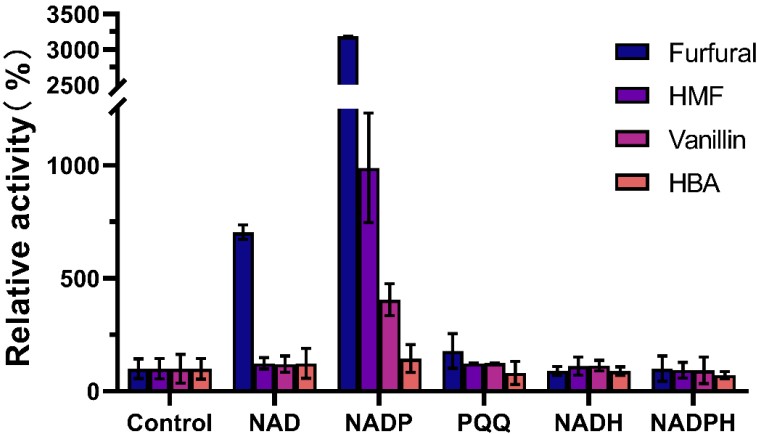

**Figure 4.** The effect of different cofactors on enzyme activity. Enzyme activity was measured by adding 1 mM NADP, 1 mM NAD, and 1 μM PQQ to the reaction system, respectively. The enzyme activity measured without cofactor was 100%.

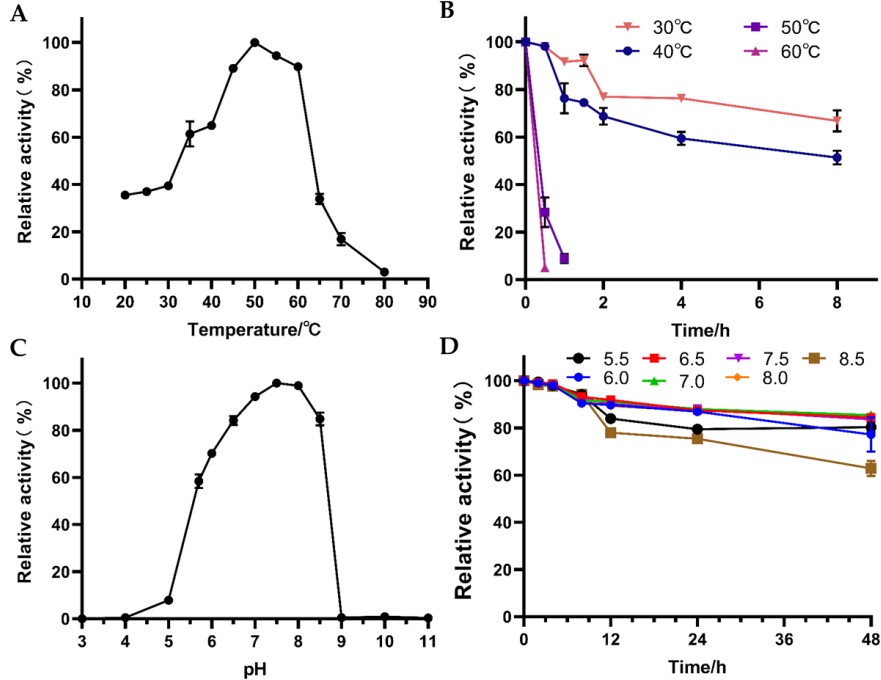

**Figure 5.** Effect of temperature and pH on enzyme activity and stability of *W826-RS0111485*. (**A**) is the effect of temperature on enzyme activity, (**B**) is the effect of temperature on enzyme stability, (**C**) is the effect of temperature on enzyme activity, (**D**) is the effect of temperature on enzyme stability. They were tested at different temperatures (20 °C, 25 °C, 30 °C, 35 °C, 40 °C, 45 °C, 50 °C, 55 °C, 60 °C, 65 °C, 70 °C, 80 °C), and different pH (pH = 4.0, 5.0, 5.5, 6.0, 6.5, 7.0, 7.5, 8.0, 8.5, 9.0) were used to measure the enzyme activity. The optimum temperature and pH of the reaction were determined to be the temperature at which the highest enzyme activity was obtained. The pure enzymes were incubated at different temperatures and pH, sampled at regular intervals, and then the residual enzyme activity was determined under the optimal reaction conditions, taking the enzyme activity without incubation as 100%.

In the process of enzymatic reaction, the pH value has a great impact on the activity and stability of the enzyme. Peracid and peralkaline can destroy the spatial structure of the enzyme and cause it to lose its activity. In this study, the enzyme activity and stability of *W826-RS0111485* at different pH values were investigated. The results showed (Figure 5C) that the optimum reaction pH of *W826-RS0111485* was 7.5, which was basically the same as the optimum pH 8.0 measured with acetaldehyde as the substrate [30]. At pH 6.5 and 8.5, the enzyme still maintained about 80% activity, but at pH 9.0, the activity decreased sharply and was basically inactive. The enzyme was pH stable and 60% of the activity remained after 48 h incubation in pH 8.5 buffer (Figure 5D). Incubation for 48 h in the pH 6.5–8.0 buffer still yielded close to 90% activity.

### 3.4. Enzyme Kinetic Parameters of W826-RS0111485 for Furfural

Furfural is an important and major inhibitor in lignocellulosic hydrolysate, especially in dilute acid pretreated hydrolysate. The previous results indicated that *W826-RS0111485* was more effective in furfural conversion, therefore enzyme kinetic analysis was performed using furfural as a substrate under optimal reaction conditions (Figure 6). The results showed that the *Km*, *Vmax*, and *Kcat* of *W826-RS0111485* to furfural were 2.45, 80.97, and 232.22 min$^{-1}$ (0.09 U/mg) (Table 1), respectively. The furfural activity of this enzyme was similar to that of GRE2 and YML131W in yeast, but the *Kcat/Km* of this enzyme was slightly higher than that of GRE2 and YML131W, indicating that the catalytic efficiency was higher than that of GRE2 and YML131W. More importantly, GRE2 and YML131W catalyzed furfural conversion to furfuryl alcohol, which required further conversion to furoic acid. However, *W826-RS0111485* could directly convert furfural to furoic acid.

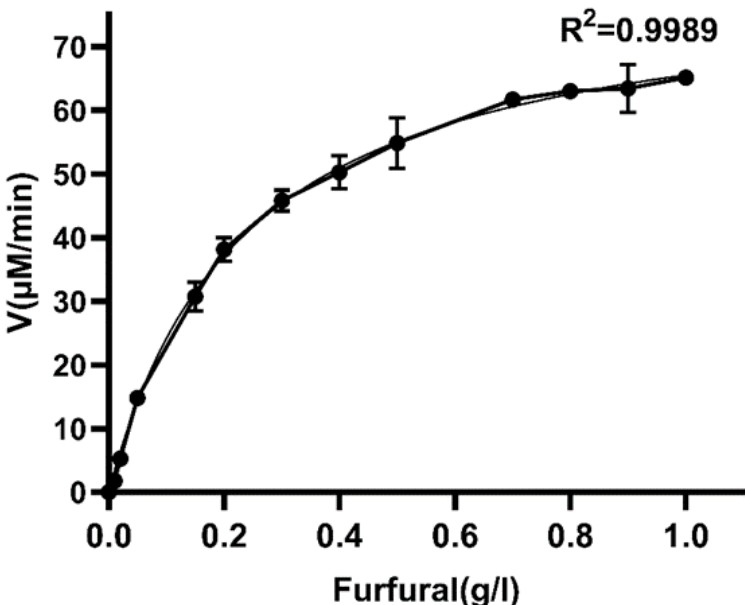

**Figure 6.** Kinetic analysis of *W826-RS0111485* for furfural.

**Table 1.** Kinetic parameters of different enzymes when furfural was used as a substrate.

| Enzyme | Km (mM) | Vmax (μM/min) | Kcat (min$^{-1}$) | Kcat/Km (mM$^{-1}$ min$^{-1}$) | Cofactor | Product | Reference |
|--------|---------|---------------|-------------------|--------------------------------|----------|---------|-----------|
| ALDH | 2.451 | 80.97 | 232.33 | 94.80 | NADP | Furoic acid | This study |
| GRE2 | 2.18 | 5.16 | 189 | 86.70 | NADP | Furfuralcohol | [31] |
| YML131W | 6.96 | 11.76 | 470.09 | 67.59 | NAD | Furfuralcohol | [20] |

### 3.5. Biodetoxification Performance of Recombinant E. coli with W826-RS0111485

The key biodetoxification genes were mined in order to improve the inhibitor conversion ability of industrial strains by means of molecular biological technology at present. In order to investigate the furfural conversion performance of this gene accurately, *E. coli* was chosen as the host stain in this study. *E. coli* containing pET28a(+)-*W826-RS0111485* recombinant plasmid was used for detoxification performance assay, and *E. coli* containing empty pET-28a(+) plasmid was used as control. The results showed that compared with the control group (Figure 7), the furfural in the experimental group could be converted into furoic acid faster. In the experimental group, the recombinant stain containing the target gene completely consumed 1 g/L furfural within 8 h. Meanwhile, the majority of furfural was converted into furoic acid and only a small amount of furfuryl alcohol was produced (Figure 7B). In the control group, the recombinant strain containing empty plasmid still had some furfural present at 96 h, and the products were basically furfuryl alcohol (Figure 7A). The target protein (*W826-RS0111485*) could significantly improve the furfural conversion performance of *E. coli*, which provided a theoretical basis for enhancing the inhibitor degradation ability of more industrial fermentation strains in the future.

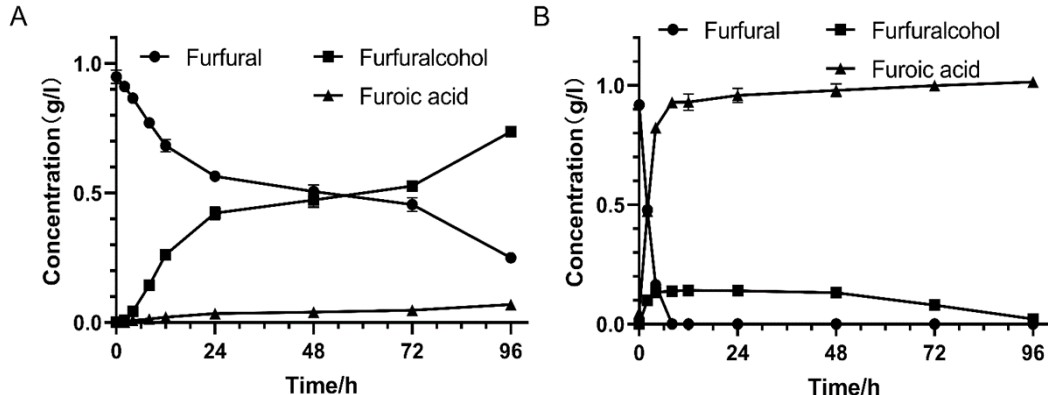

**Figure 7.** Degradation of furfural by recombinant *E. coli* with *W826-RS0111485*. (**A**) is the control group, and *E. coli* contains empty pET-28a(+) plasmid. (**B**) is the experimental group, and *E. coli* contains pET28a(+)-*W826-RS0111485* recombinant plasmid. *E. coli* containing empty plasmid pET-28a(+) and recombinant plasmid pET-28a(+)-*W826-RS0111485* were induced overnight, centrifuged at $7000 \times g$ for 10 min, and resuspended in 0.02 M, pH 7.4 phosphate buffer. The furfural was added to the final concentration of 1 g/L, and the cell concentration was set to $1 \times 10^9$, that is, the OD600nm was about 1.0. The cells were shaken at 37 °C and 220 rpm. Samples were taken at regular intervals and centrifuged at 12,000 rpm for 10 min, and the supernatant was used to detect inhibitor degradation.

### 4. Conclusions

A novel gene *W826-RS0111485*, associated with inhibitor degradation, was mined from *G. oxydans* DSM 2003 by protein profiling. It was heterologously expressed in *E. coli* to an aldehyde dehydrogenase, which could directly convert high toxic aldehyde inhibitors to weak toxic acids, and in particular, convert furfural to furoic acid efficiently. For this aldehyde dehydrogenase, the most suitable cofactor was NADP when aldehyde inhibitor was the substrate. The optimal temperature was 50 °C, the optimal pH was 7.5, and the *Km*, *Vmax*, and *Kcat* under furfural stress were 2.45, 80.97, and 232.22 min$^{-1}$, respectively. The furfural degradation performance of recombinant *E. coli* containing the target gene was significantly increased, which indicated that the *W826-RS0111485* was an excellent candidate gene for detoxification. This study provided a theoretical basis for enhancing the inhibitors degradation ability of more industrial strains. In this way, detoxification process will be weakened or reduced, production costs will be effectively reduced, and the vigorous development of biorefinery field will be promoted in the future.

**Author Contributions:** H.Z.: Investigation, Writing-original draft; J.J.: Investigation, Protein expression and purification; C.Q.: Enzymatic characteristics; G.Z.: Sequence analysis; G.M.: Writing-review and editing, Supervision; H.X.: Writing-review and editing, Supervision; F.W.: Writing-review and editing, Supervision; Z.W.: Supervision; J.Z.: Writing-review and editing, Supervision; P.Z.: Investigation, Writing-original draft, Supervision; A.S.: Supervision. All authors have read and agreed to the published version of the manuscript.

**Funding:** This work was supported by the National Natural Science Foundation of China (21908044), (22178105), the Key Scientific Research Project of Universities of Henan Province (21B180005) and the Key Research and Development Foundation of Henan (202102110299).

**Institutional Review Board Statement:** Not applicable.

**Informed Consent Statement:** Not applicable.

**Data Availability Statement:** All data generated or analyzed during this study are included in this published article.

**Conflicts of Interest:** The authors declare that they have no known competing financial interests or personal relationships that could have appeared to influence the work reported in this paper.

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
