# Peer review of "Identification of a Novel Dehydrogenase from Gluconobacter oxydans for Degradation of Inhibitors Derived from Lignocellulosic Biomass"

_fermentation, doi:10.3390/fermentation9030286_

Round 1
Reviewer 1 Report
The manuscript possesses a high level of scientific potential
It is well-written and easy to comprehend.
The results are presented in a suitable form and are discussed properly
The conclusion is in accordance with the results
The manuscript can be published in its present form, without corrections
Author Response
Thank you for your comments on our paper " Identification of A Novel Dehydrogenase From Gluconobacter Oxydans For Degradation of Inhibitors Derived From Lignocellulosic Biomass", we appreciate your comments about our manuscript.
Reviewer 2 Report
In the present article, the authors identified an aldehyde dehydrogenase gene W826- RS0111485 by protein profiling, then a series of enzymatic properties were thereafter determined. The topic and results are interesting, and the conclusion summarizes the main points. Minor revisions are required before publication.
(1) Many thanks to the author for investigating the impact of the pH value on the activity and stability of the enzyme. Very commending and impactful.
(2) There are some typos in the report. Please check the entire manuscript for typos and correct the wrong grammar.
Author Response
Point 1: Many thanks to the author for investigating the impact of the pH value on the activity and stability of the enzyme. Very commending and impactful.
Response 1: Thank you for your recognition of this manuscript.
Point 2: There are some typos in the report. Please check the entire manuscript for typos and correct the wrong grammar.
Response 2: We are deeply sorry for the spelling and grammar mistakes caused by our carelessness. We made a great effort to revise the article, remove errors, and improve the writing. For example, in line 54, “The results of a recent study show that…” was revised to “The results of a recent study showed that…”; in line 142, “2.5. Protein mass spectrometry” was revised to “2.6. Protein mass spectrometry”. We have corrected these errors in the revised manuscript and highlighted them in red. Your suggestions will be of great help to us in polishing the manuscript.

Reviewer 3 Report
The manuscript “Identification of a novel dehydrogenase from Gluconobacter oxydans for degradation of inhibitors derived from lignocel- 3 lulosic biomas” presents important results when describing the degradation performance of inhibitors derived from lignocellulosic biomass by a new recombinant E. coli. The study has a relevant contribution to applications in biorefinery processes.
Below are comments on the article: - In item 2.1, describe the origin of the tested Gluconobacter oxydans isolate, in addition to the time and temperature at which it was cultivated. - In item 2.2, mention the bibliographical references of the tested methodology, in addition to the brand of enzymes used. - In experiments on enzyme activity and stability, why choose such a wide temperature range? In which temperature range do degradation reactions of lignocellulosic biomass usually occur? - In item 2.5 Inhibitor Analytical methods: detail the conditions tested and cite the bibliographic references. Other Item 2.5 Protein mass spectrometry: Review the numbering. Describe in detail how the enzymatic digestion was performed and the analysis performed using LCMSMS, in addition to defining the acronym. - In the conclusion, mention what would be the future prospects for carrying out the experiments on a biorefinery scale.
Author Response
Thank you for your comments and these comments are very helpful for us to revise and improve our manuscript. We have made efforts to revise the manuscript and marked the modified parts in red. We greatly appreciate your comments and suggestions.
Point 1: In item 2.1, describe the origin of the tested Gluconobacter oxydans isolate, in addition to the time and temperature at which it was cultivated.
Response 1: The origin of the tested Gluconobacter oxydans isolate was described in line 93-94, “G. oxydans DSM 2003 used in this study was purchased from German Collection of Microorganisms and Cell Cultures, Germany, Braunschweig.” The time and temperature were added in revised manuscript, “The strains were cultured in seed medium containing 80 g/L sorbitol, 20 g/L yeast extract, 0.5 g/L MgCl2·7H2O, 1.5 g/L KH2PO4 and 1.5 g/L (NH4)2SO4. The seed medium was sterilized at 115 °C for 15 min. The culture conditions were 30 °C, 220 rpm for 24 hours, inoculum size was 10%.”
Point 2: In item 2.2, mention the bibliographical references of the tested methodology, in addition to the brand of enzymes used.
Response 2: The enzyme digestion method used in this study was referred to the literature 19 (Mao et al.,2021). The restriction enzyme was from the New England Biolabs, Beijing, China. The corresponding content has been revised in the manuscript and highlighted in red.
Point 3: In experiments on enzyme activity and stability, why choose such a wide temperature range? In which temperature range do degradation reactions of lignocellulosic biomass usually occur?
Response 3: First, the growth temperature of G. oxydans DSM2003 was 30℃, and the enzymatic hydrolysis temperature of lignocellulosic biomass was about 55℃. Therefore, in this study, the temperature range was set around the optimal growth temperature of G. oxydans DSM2003 and the enzymatic hydrolysis temperature of lignocellulosic biomass.
Point 4: In item 2.5 Inhibitor Analytical methods: detail the conditions tested and cite the bibliographic references.
Response 4: The determination methods of inhibitors and their metabolites refer to the literature 16 (Zhou et al., 2019). The corresponding content has been revised in the manuscript and highlighted in red.
Point 5: Other Item 2.5 Protein mass spectrometry: Review the numbering. Describe in detail how the enzymatic digestion was performed and the analysis performed using LCMSMS, in addition to defining the acronym.
Response 5: Firstly, the item “2.5 Protein mass spectrometry” was revised to “2.6. Protein mass spectrometry”. In this study, the Protein mass spectrometry was submitted to Shanghai Applied Protein Technology Co. Ltd. The QE mass spectrometer was used to obtain tandem mass spectra by the principle of HCD (Higher-energy collisional dissociation). The detailed method was added in line 146-157, “The detail method used in this study was as follows: (1) Enzymolysis of protein samples. After reduction and alkylation, the protein samples were added with Trypsin, and the enzymolysis was conducted at 37℃ for 20 h. The enzymatic hydrolysis products were freeze-dried after desalting, redissolved in 0.1% FA solution and stored at -20℃ for future use. (2) Mass spectrometry. Liquid A was 0.1% formic acid aqueous solution, and liquid B was 0.1% formic acid acetonitrile aqueous solution (84% acetonitrile). After the column was balanced with 95% liquid A, samples were fed to the Trap column by an automatic sampler. (3) Mass spectrum data collection. Mass charge ratios of peptides and peptide fragments were collected as follows: 20 fragment maps (MS2 scan) were collected after each full scan. (4) Data analysis. The raw file of mass spectrometry test was retrieved from the corresponding data-base with Proteome Discoverer1.4 software, and finally the identified protein results were obtained.”
Point 6: In the conclusion, mention what would be the future prospects for carrying out the experiments on a biorefinery scale.
Response 6: In the conclusion, the relevant content has been added in line 329-334, “The furfural degradation performance of recombinant E. coli containing the target gene was significantly increased, which indicated that the W826-RS0111485 was an excellent candidate gene for detoxification. This study provided a theoretical basis for enhancing the inhibitors degradation ability of more industrial strains. In this way, detoxification process will be weakened or reduced, production costs will be effectively reduced, and the vigorous development of biorefinery field will be promoted in the future.”
